# Antimicrobial Activity of Chemically and Biologically Treated Chitosan Prepared from Black Soldier Fly (*Hermetia illucens*) Pupal Shell Waste

**DOI:** 10.3390/microorganisms9122417

**Published:** 2021-11-23

**Authors:** Mevin Kiprotich Lagat, Samuel Were, Francis Ndwigah, Violah Jepkogei Kemboi, Carolyne Kipkoech, Chrysantus Mbi Tanga

**Affiliations:** 1Department of Botany, Jomo Kenyatta University of Agriculture and Technology, Nairobi P.O. Box 62000-02000, Kenya; mlagat237@gmail.com (M.K.L.); samaringo@gmail.com (S.W.); irerifin@gmail.com (F.N.); violahkemboi42@gmail.com (V.J.K.); 2Department of Food and Nutritional Sciences, Jomo Kenyatta University of Agriculture and Technology, Nairobi P.O. Box 62000-02000, Kenya; 3International Centre of Insect Physiology and Ecology (*icipe*), Nairobi P.O. Box 30772-00100, Kenya; ctanga@icipe.org

**Keywords:** insects, *Hermetia illucens*, pupal exuviae, chitin, chitosan, antimicrobial activity keyword

## Abstract

Globally, the broad-spectrum antimicrobial activity of chitin and chitosan has been widely documented. However, very little research attention has focused on chitin and chitosan extracted from black soldier fly pupal exuviae, which are abundantly present as byproducts from insect-farming enterprises. This study presents the first comparative analysis of chemical and biological extraction of chitin and chitosan from BSF pupal exuviae. The antibacterial activity of chitosan was also evaluated. For chemical extraction, demineralization and deproteinization were carried out using 1 M hydrochloric acid at 100 °C for 2 h and 1 M NaOH for 4 h at 100 °C, respectively. Biological chitin extraction was carried out by protease-producing bacteria and lactic-acid-producing bacteria for protein and mineral removal, respectively. The extracted chitin was converted to chitosan via deacetylation using 40% NaOH for 8 h at 100 °C. Chitin characterization was done using FTIR spectroscopy, while the antimicrobial properties were determined using the disc diffusion method. Chemical and biological extraction gave a chitin yield of 10.18% and 11.85%, respectively. A maximum chitosan yield of 6.58% was achieved via chemical treatment. From the FTIR results, biological and chemical chitin showed characteristic chitin peaks at 1650 and 1550 cm^−1^—wavenumbers corresponding to amide I stretching and amide II bending, respectively. There was significant growth inhibition for *Escherichia coli*, *Bacillus subtilis**,*
*Pseudomonas aeruginosa**,*
*Staphylococcus aureus*, and *Candida albicans* when subjected to 2.5 and 5% concentrations of chitosan. Our findings demonstrate that chitosan from BSF pupal exuviae could be a promising and novel therapeutic agent for drug development against resistant strains of bacteria.

## 1. Introduction

Chitin is a natural polysaccharide that exists as the second most common polymer after cellulose [1]. It occurs as a structural component in most crustacean shells, the pupal exuviae of some insects, and in the cell membranes of fungi [2]. Currently, chitin and chitosan have garnered attention for extensive applications in wastewater treatment, the pharmaceutical industry, food manufacturing, and agriculture [3,4,5,6]. This is because they possess various unique properties, including biocompatibility, biodegradability, non-toxicity, and antimicrobial activities [7].

Extraction of chitin and chitosan can be performed using chemical and biological methods [8]. The process involves three steps: deproteinization, demineralization, and deacetylation for the conversion of chitin to chitosan. The chemical method uses acids and bases such as hydrochloric acid and sodium hydroxide for the removal of minerals and proteins, respectively [9]. High concentrations of these chemicals severely pollute the environment, affect the physicochemical properties of chitin and chitosan, and are also expensive [10]. To overcome these challenges, there is an alternative to using biological chitin extraction methods, involving the utilization of microorganisms such as lactic-acid-producing bacteria for demineralization and proteolytic bacteria for deproteinization. The deacetylase enzyme is then used for the biological conversion of chitin to chitosan [11]. Biological extraction of chitin offers high reproducibility, is simpler to manipulate, requires a small number of solvents—making it cheaper—and is eco-friendly, with a lower energy input.

Several studies have focused on chitin extracted from arthropods such as millipedes, beetles, and grasshoppers [12,13,14], as well as from mushroom species such as *Pleurotus ostreatus*, *Ganoderma lucidum*, and *Agaricus bisporus* [15,16,17]. However, not much work has been done on the black soldier fly, *Hermetia illucens* L. To the best of our knowledge, this is the first report on the biological extraction of chitin from BSF pupal exuviae (i.e., a byproduct abundantly available and highly accessible year-round in commercial production facilities worldwide) using bacteria. Herein, the percentage yield of chitin and chitosan was determined before characterization using Fourier-transform infrared spectra and scanning electron microscopy. The efficiency of biological and chemical extraction methods was established to understand the economic viability of both methods. Finally, the antimicrobial activities of chitosan against selected clinical pathogenic bacteria were tested.

## 2. Materials and Methods

### 2.1. Study Site

This research was conducted at the Jomo Kenyatta University of Agriculture and Technology (JKUAT), 30 km northeast of Nairobi metropolis, at the Government of Kenya (GoK) Botany Laboratory (10180S,370E). The pupal exuviae from which both chitin and chitosan were extracted was obtained from the BSF mass production facility at the Animal Rearing and Containment Unit (ARCU) at the International Centre of Insect Physiology and Ecology (*icipe*), Nairobi, Kenya.

### 2.2. Chemical Extraction of Chitin from Hermetia illucens Pupal Exuviae

The BSFs were reared following the standard operating procedures at *icipe* [18]. The samples were collected after three independent rearing periods in April, July, and September 2020. The pupal exuviae were sorted, washed in water, and blended into a fine powder using a commercial laboratory blender (Sanyo SM-1250 GC, Zhongshan, China). One hundred grams of the resultant powder was weighed using an analytical balance (RADWAG, model WTB 2000, Radom, Poland), and treated with 1000 mL of 1 M sodium hydroxide (Sigma-Aldrich, St. Louis, MI, USA) and allowed to boil for 4 h, with continuous stirring to remove proteins. The product was thoroughly washed with distilled water until reaching a neutral pH. The pH was measured using the HANNA HI 2211 pH/ORP Meter (Helsinki, Finland). The product was then dried at a temperature of 60 °C for 24 h using a hot-air oven (GRANDE GMH-225, Dongguan, China). Calcium carbonate was removed from the dried sample by treating it with 1000 mL of 1 M hydrochloric acid (Sigma-Aldrich, MI, USA) and allowing it to boil for 2 h. The final product was rinsed in distilled water until a neutral pH was achieved; it was then dried in a hot-air oven at a temperature of 60 °C for 24 h. The final dried product was weighed, and the percentage yield of chitin was calculated from the initial pupal exuviae used, following the method used in [19]. All procedures were carried out in triplicate.

### 2.3. Biological Extraction of Chitin from Hermetia illucens Pupal Exuviae

#### 2.3.1. Bacteria Isolates

Biological extraction of chitin was carried out using protease-producing bacteria (*Bacillus subtilis* and *Pseudomonas aeruginosa)* and lactic-acid-producing bacteria (*Lactobacillus plantarum)* following the method described in [20], with slight modifications. The isolates used were obtained from the JKUAT food microbiology and botany laboratories. These isolates were used because they have been reported to give high yields in chitin extraction [8].

#### 2.3.2. Biological Extraction

There were two sets of experiments: In the first setup, deproteinization was performed using proteolytic bacteria—namely, *Bacillus subtilis* and *Pseudomonas aeruginosa*—followed by demineralization using *Lactobacillus plantarum*. In the second series of experiments, demineralization was performed followed by deproteinization. Each of the sets of experiments was replicated three times to increase the reliability and accuracy of the results obtained, and repeated for the three different independent samples.

#### 2.3.3. Deproteinization and Demineralization

During the deproteinization process, 100 g of the pupal exuviae flour was soaked in 2 L of distilled water. To this mix, 55 g of sucrose (Sigma-Aldrich, St. Louis, MI, USA) was added, and the mixture was then inoculated with 10 mL of *Bacillus subtilis* and *Pseudomonas aeruginosa* and incubated at 37 °C for 5 days. After 5 days, the deproteinized material was sterilized by autoclaving for 15 min at 121 °C using an IKENOHATA AL-300 autoclave (Sibata Scientific Technology Ltd., Tokyo, Japan), and then demineralized by inoculating it with 10 mL of *Lactobacillus plantarum* and incubating it at 37 °C in anaerobic conditions for another 5 days [21].

For co-cultivation of *B. subtilis* and *P. aeruginosa* in chitin extraction, 100 g of the pupal exuviae powder was soaked in 2 L of distilled water. To this mix, 55 g of sucrose (Sigma-Aldrich, MI, USA) was added, and the mixture was then inoculated with 10 mL of *B. subtilis* and *P. aeruginosa*, and incubated at 37 °C for 5 days using a JSI 102 incubator Patna, India). After 5 days, the deproteinized material was sterilized and then demineralized by inoculating it with 10 mL of *L. plantarum*, and incubated at 37 °C in anaerobic conditions for another 5 days.

### 2.4. Deacetylation of Chitin to Chitosan

The extracted chitin from *H. illucens* pupal exuviae was then converted into chitosan by refluxing 200 g in 1000 mL of 40% NaOH (Sigma-Aldrich, St. Louis, MI, USA) solution and boiling for 8 h. The deacetylated chitosan was then cleaned with sterile water to neutral pH and dried in a hot-air oven at 60 °C. The dried samples were then stored at 4 °C in airtight plastic containers until use.

### 2.5. Determination of Percentage Yield of Chitin and Chitosan

The chitin yield was calculated based on the dry weight using gravimetric measurements between the raw pupal shell waste and the chitin that was obtained after the extraction process. This was done after drying the pupal exuviae, chitin, and chitosan to a constant weight to ensure that the difference in weight was not attributed to the moisture content. Furthermore, the chitosan yield was calculated based on the dry weight from the weight difference between the chitin that was obtained and the chitosan that was obtained after the chitosan production process, using the following equations:(1)$$\mathrm{Chitin}\;\mathrm{yield}\;(\%)=\frac{a}{b}\times 100$$
where **a** is the obtained chitin weight (g) and **b** is the pupal shell waste weight (g).
(2)$$\mathrm{Chitosan}\;\mathrm{yield}\;(\%)=\frac{c}{d}\times 100$$
where **c** is the weight (g) of chitosan that was obtained, while **d** is the weight (g) of chitin that was prepared.

### 2.6. Characterization of Chitin and Chitosan by Fourier-Transform Infrared (FTIR) Spectroscopy

The FTIR measurements were done following the method described in [22], using FTIR with a scanning range of 4000 to 400 cm^−1^ (BRUCKER, model ALPHA, Berlin, Germany).

### 2.7. Characterisation of Chitin and Chitosan by Scanning Electron Microscopy (SEM)

The surface morphologies of chitosan and chitin samples were determined using a JCM-7000 NeoScope Benchtop scanning electron microscope (JEOL Ltd., Tokyo, Japan). The samples were coated with carbon film and examined using the secondary electron mode with an accelerating voltage of 15 Kv to show microstructures at different magnifications.

### 2.8. Antimicrobial Activities

#### 2.8.1. Test Organisms for Antimicrobial Assay

The antimicrobial activities of chitosan were tested against two Gram-negative bacteria—*Escherichia coli* ATCC^®^25922 (*E. coli*) and *Pseudomonas aeruginosa* ATCC^®^27853 *(P. aeruginosa*)—two Gram-positive bacteria—*Staphylococcus aureus* ATCC^®^25923 (*S. aureus*) and *Bacillus subtilis* ATCC^®^11778 *(B. subtilis*)—and the yeast *Candida albicans* ATCC^®^10231 (*C. albicans*). The test microbes were obtained from medical microbiology and Government of Kenya (GoK) laboratories at JKUAT.

#### 2.8.2. Inoculum and Sample Preparation

Each bacterial strain was sub-cultured overnight at 37 °C in nutrient broth (Sigma-Aldrich, St. Louis, MI, USA), while *C. albicans* was sub-cultured overnight at 37 °C in Sabouraud dextrose broth (Sigma-Aldrich, MI, USA), and then on agar plates. Bacterial colonies were thereafter suspended in 9 mL of sterile saline solution, and the suspension was adjusted to achieve turbidity equivalent to the 0.5 McFarland standard, which is equivalent to 1 × 10^8^ colony-forming units (CFUs)/mL. Mueller–Hinton agar (Sigma-Aldrich, St. Louis, MI, USA) was then prepared, and 12 ml of the medium was dispensed to the Petri dishes and allowed to solidify.

#### 2.8.3. Antimicrobial Susceptibility Assay

The Kirby–Bauer disc diffusion method was employed in this study for antimicrobial assay [23]. Typically, 20 µL of freshly prepared McFarland bacterial cultures of *E. coli*, *S. aureus*, *P. aeruginosa*, and *B subtilis*, as well as a fungal culture of *C. albicans*, were then inoculated and spread uniformly onto Mueller–Hinton agar plates. Chitosan sample discs—prepared by impregnating 50 µL (250 µg/disc) of chitosan solution on sterile filter paper discs (6 mm), followed by air-drying—were placed on the top of the agar plates. The plates were then incubated at 37 °C for 24 h using an IS62 incubator (Genlab, Tokyo, Japan). The presence of inhibition zones was measured around each disc in millimeters (mm) and was considered as evidence of antimicrobial activity. The experiments for each test organism were carried out in triplicate. Filter-paper discs soaked in 2 mL of 1% acetic acid (Sigma-Aldrich, MI, USA) without chitosan were used as a positive control, while sterile distilled water was used as a negative control. The effects of BSF chitosan were also compared to a group of standard reference antibiotics (Alpha Medical Manufacturers, Nairobi, Kenya)—including ampicillin (25 mcg), streptomycin (25 mcg), tetracycline (100 mcg), nalidixic acid (30 mcg), chloramphenicol (25 mcg), clotrimazole (25 mcg), and nitrofurantoin (200 mcg)—to ascertain whether chitosan can be used as an antibiotic compound.

### 2.9. Data Collection and Analysis

Data on chitin yield were collected based on the dry weight using gravimetric measurements between the raw pupal shell waste and the chitin that was obtained after the extraction process. Data on the chitosan yield were collected based on the dry weight from the weight difference between the chitin that was obtained and the chitosan that was obtained after the chitosan production process. The data on antimicrobial activity were collected by measuring clear zones of inhibition around each disc on each plate of each test organism. Data analysis was done using Stata SE-64 2011 statistics software, and means were separated using the Bonferroni range test. Data on the chitin and chitosan yields were expressed as mean ± standard deviation, and comparisons of mean yield between groups were performed by one-way analysis of variance. Similarly, the difference in the mean inhibition zone was measured using a one-way analysis of variance, and the difference was considered significant at *p* ≤ 0.05.

## 3. Results and Discussions

### 3.1. Chitin Yield from the Pupal Exuviae of the Black Soldier Fly

The chitin yield obtained from co-cultivation of bacteria was significantly (*p* > 0.05) higher compared to each of the individual cultivated bacteria (Table 1). There was, however, a significant difference in the yields obtained from chemical and biological extraction methods with individual fermentation. Co-cultivation of bacteria gave the highest yield—almost equal to the chemical extraction. In individual fermentation, *P. aeruginosa* gave the highest yield, while the lowest yield was obtained by *B. subtilis*. When the reverse order of extraction was employed (demineralization then deproteinization), there was an increase in the yield of chitin, with *B. subtilis* giving a higher yield than *P. aeruginosa*.

The chitin yield obtained from co-cultivation of bacteria was significantly (*p* > 0.05) higher compared to each of the individual cultivated bacteria (Table 1). There was, however, a significant difference in the yields obtained from chemical and biological extraction methods with individual fermentation. Co-cultivation of bacteria gave the highest yield—almost equal to the chemical extraction. In individual fermentation, *P. aeruginosa* gave the highest yield, while the lowest yield was obtained by *B. subtilis*. When the reverse order of extraction was employed (demineralization then deproteinization), there was an increase in the yield of chitin, with *B. subtilis* giving a higher yield than *P. aeruginosa*.

The present findings on the yield obtained from both biological and chemical extraction differ from the findings reported by the authors of [24,25,26], who obtained slightly lower yields. This difference could be due to the use of pupal exuviae in the present study, while the previous studies used dead flies [27]. Our findings are consistent with those reported by the author of [19], who demonstrated that the sequence of extraction, deproteinization (DP), and demineralization (DM) can lead to a decrease in chitin yield. This is because DP before DM erodes the protein layer that covers the chitin matrix, exposing it to acidic treatment and causing significant removal of inorganic material. Significant hydrolysis and loss of chitin fraction resulted in a low yield of chitin [19]. The chitin yield obtained in this study via biological means differed from the results documented in [28], where the authors found a higher chitin yield from crabs [28]. This variation was due to different sources, such as marine organisms, fungi, and insects [29]. The results obtained from this study can be compared with the yield obtained from other insects—for example, grasshoppers, beetles, and *Daphnia* sp. [30,31,32]. In this study, the chitosan yield obtained from the BSF pupal exuviae (6.58%) via chemical means was lower when compared to that reported in [19]. According to the findings of [19], the variation in chitosan yield may have been due to excessive depolymerization of the chitosan polymer, and loss of the sample due to excessive removal of acetyl groups during deacetylation. The chitosan yield obtained was also comparable to other sources, such as cockroaches, shrimp, *Musca domestica,* and krill [33,34,35].

### 3.2. Analysis of Functional Groups in Chitin Extracted from the Black Soldier Fly

There were similarities observed in the structure of chitin extracted through biological and chemical methods (Figure 1). Biologically extracted chitin exhibited vibration peaks at 3440, 2355, 1650, and 1550 cm^−1^, while chemically extracted chitin showed vibration peaks at 3440, 2355, 1651, and 1552 cm^−1^.

Commercial shrimp chitin had peaks at 3440, 2355, 1652, and 1552 cm^−1^. The band at 3440 cm^−1^ was broad and was due to the presence of hydroxyl groups in the chitin. The band at 2355 cm^−1^ was attributed to the presence of C–H stretching vibrations, which were indicative of the presence of methyl groups. Additionally, the band at 1650 cm^−1^ was attributed to C=O secondary amide I stretching vibrations, while the band at 1550 cm^−1^ was attributed to amide II, which was indicative of N-H bending.

These findings were similar to those in [19], where the author used chitin from BSF larvae and adult flies, and obtained the same characteristic peaks. These results are also consistent with the findings of [30], where the authors used chitin from grasshoppers, and obtained the same characteristic bands [19,36]. This study revealed that the chitin extracted from the BSF pupal exuviae via biological and chemical means is of α-form, and is very similar to that of commercial shrimp chitin. In the literature, it is reported that alpha (α)-chitin has characteristic chitin peaks recorded at 1650, 1620, and 1550 cm^−1^ [30].

In addition, [37] found that the presence of the glycosidic bond at 896 cm^−1^ is a characteristic band for alpha-chitin, which was detected in all chitin samples (Figure 1). It was found that there was no difference between the structures of biological, chemical, and shrimp chitin. In all of the chitin samples, there were no peaks at 1540 cm^−1^. The absence of bands at 1540 cm^−1^ could be attributed to the absence of protein contaminants showing sufficient deproteinization [17,37].

Other major bands detected were 1413 cm^−1^ (CH_2_ ending and CH_3_ deformation), 1256 cm^−1^ (C–H bending and CH_3_ symmetrical deformation), 1069 cm^−1^ (asymmetric in-phase ring-stretching mode), 1023 cm^−1^ (C-O-C asymmetric in-phase ring-stretching), 1159 cm^−1^ (CH_2_ wagging), 1114 cm^−1^ (asymmetric bridge oxygen stretching), and 896 cm^−1^ (glycosidic bond). The results obtained were consistent with the results of previous studies [19].

### 3.3. Analysis of Functional Groups in Chitosan Extracted from Black Soldier Fly

The characteristic bands were recorded at 1650 cm^−1^ and 1587 cm^−1^ for the chitosan from BSF, and at 1649 cm^−^^1^ and 1587 cm^−1^ for commercial chitosan.

The characteristic peaks observed at 1650 cm^−1^ and 1649 cm^−1^ in both BSF and standard commercial chitosan were due to the presence of amide I (C=O) in the acetamide group (NHCOCH_3_), while the peaks observed at 1587 cm^−1^ in both chitosan samples were due to the amide II band (NH_2_) in the NHCOCH_3_ group. These present findings are consistent with those reported in [19], where the author reported similar characteristic peaks using chitosan from BSF larvae and adult flies. The findings of [19] showed that the peaks at around 1650–1655 cm^−1^ and 1583–1590 cm^−1^, which correspond to (C=O) in the NHCOCH_3_ group (amide I band) and (NH_2_) in the NHCOCH_3_ group (amide II band), respectively, were characteristic of chitosan.

Additional broad absorption bands observed at 3250–3750 cm^−1^ were attributed to symmetric stretching vibrations of the O-H and NH_2_ groups caused by the strong intermolecular hydrogen bonding of chitosan polysaccharides. The peaks at approximately 2350 cm^−1^ were attributed to symmetric and asymmetric vibrations of C-H groups in the chitosan samples. The absorption peaks at 1374 and 1255 cm^−1^ were ascribed to N-H bending vibrations of primary amides and C-O-C stretching vibrations in both chitosan samples. These findings were consistent with those obtained in [37], where the authors used chitosan derived from grasshopper species. The peaks displayed at around 1153 cm^−1^ and 1082 cm^−1^ were attributed to the β (1–4) glycosidic bond in the polysaccharide unit and the stretching vibrations of C-O-C in the glucose ring, respectively. These findings are similar to those found in [38], where the authors used chitosan from shrimp shells.

The absorption band of amide II had a lower intensity than that of amide I, suggesting effective deacetylation. When chitin was converted to chitosan, the intensity of the amide II absorption band decreased, while the intensity of amide II increased, showing the formation of amide (NH_2_) groups. The FTIR results suggest that there was a similarity between the chemical composition and the bonding types of chitosan in the pupal shells and commercial shrimp chitosan (Figure 2). These findings are similar to those reported in [39], where the authors used chitosan from nymphs and adult grasshoppers.

### 3.4. Characterization of Chitin and Chitosan by Scanning Electron Microscopy (SEM)

The surface morphology of chitin obtained from BSF pupal exuviae by *B. subtilis*, *P. aeruginosa*, and their co-cultivation showed a rough, tightly packed structure with repeating circular and hexagonal units in a honeycomb-like arrangement (Figure 3a–c). The biologically extracted chitin had rough surfaces, with numerous porous fibers. The pores were 50 µm in diameter at X500 magnification. An increase in magnification to X1000 revealed that all of the biologically extracted chitin had fibers and pores, but low magnification gave clearer micrographs. These findings are similar to previous research on BSF pupal shell chitin reported in [19], where the surface morphology revealed a tightly packed structure with repeating circular and hexagonal units in a honeycomb-like arrangement; the author observed a rough surface with fibers and pores with a diameter of 20 µm [19].

On the other hand, chitin obtained via chemical means had a smooth surface with few fibers and pores, and the surface morphology revealed tightly packed structures with repeating circular and hexagonal units in a honeycomb-like arrangement (Figure 3d). These observations are similar to those of previous studies on BSF pupae chitin [19,30]. Commercial shrimp chitin had a rocky smooth surface, without fibers, and numerous pores of 100 µm in diameter at X500 magnification (Figure 3e), which was similar to the findings of previous studies [19,31,40].

The BSF pupal exuviae chitosan obtained via chemical means had a rough surface, without any fibers and/or pores (Figure 4a), while commercial shrimp chitosan had a dense, smooth surface without fibers or pores (Figure 4b). The findings on the morphology of chitosan extracted from BSF pupal exuviae are consistent with previous studies done on BSF chitosan from pupal shells [19,36]. Commercial shrimp chitosan, on the other hand, had a dense, smooth surface without fibers and/or pores; this finding differs from the observations of previous studies conducted by several authors [19,31,40].

According to the literature, chitin obtained from various sources has variable morphology [41]. The surface morphology of the biologically extracted chitin from the BSF pupal exuviae in this study was similar to that of other organisms, such as crabs, grasshoppers, potato beetles, locusts, and house crickets [31,40,41,42,43].

The surface morphology of chemically extracted chitin from the BSF pupal exuviae was similar to that of other sources—for instance, fungal sources such as *L. vellereus* and *P. ribis* [39], and lichen species such as *X. parietina* [44].

Biologically and chemically extracted chitin from BSF pupal shells were similar in that chitin obtained using both techniques had rough surface morphology, the same shape, and both had pores and fibers, but biologically extracted chitin had more fibers and pores; both differed from commercial shrimp chitin in that the latter had a smooth surface, a rocky shape, and numerous pores, but no fibers. The surface morphology of chitin is critical in determining its use in different fields of application, which is consistent with the findings of many researchers [19,41,45]. For example, chitin with a porous structure has been employed in the absorption of toxic metal ions, used in controlled drug delivery and tissue engineering, while chitin with a fibrous structure is used in textiles [46].

### 3.5. Antimicrobial Properties of Chitosan against Pathogenic Microbes

The concentrations of BSF chitosan (0.5, 1, 2.5, and 5%) exhibited statistically significant differences in their activity against all of the tested organisms. Chitosan from the BSF pupal exuviae exhibited varying degrees of antibacterial activity against *E. coli*, *S. aureus*, *P. aeruginosa*, *B. subtilis*, and *C. albicans*. One percent (1%) acetic acid without chitosan (positive control) demonstrated antimicrobial activity against all of the tested microbes, while the negative control (distilled water) showed no inhibitory effects (Table 2).

The inhibitory activity against each of the tested microbes increased with the increase in the concentration of chitosan. The chitosan from BSF pupal exuviae showed the largest inhibition zone against *C. albicans*. These findings contrast with the findings of [47,48], which reported a small inhibition zone when chitosan from grasshopper species and BSF were used, respectively. In this study, chitosan inhibited the growth of *E.coli*, and these findings are similar to those of previous studies that tested chitosan’s activity on *E. coli* and found a diameter of 10 mm [49,50]. Chitosan from *H. illucens* showed stronger antibacterial activity against *S. aureus* as compared to the findings of [47,51,52], where the inhibition zones were 14 and 18–21 mm for *S. aureus*, respectively. On the other hand, a larger diameter of inhibition of 33 mm has been reported based on their findings for *S. aureus* [53]. The BSF chitosan demonstrated stronger activity against *P. aeruginosa* and gave a diameter of inhibition of 21 mm, which was lower compared to the findings of [53,54], with diameters of 27 mm and 12 mm, respectively [53,54]. Chitosan from BSF demonstrated notable antibacterial activity against *B. subtilis*, which was consistent with previous studies [55]. The chitosan of BSF also showed stronger antifungal properties, as demonstrated by greater inhibition of *C. albicans*; this is similar to the findings of [47,56,57]. The activity of chitosan on *C. albicans* was higher than in [53]. From the results obtained, there is a possibility that chitosan derived from BSF has a higher antimicrobial quality when compared to chitosan derived from fungi or shrimp.

### 3.6. Comparative Effects of the Antimicrobial Activity of Chitosan from BSF, Shrimp, and Conventional Antibiotics against Selected Pathogenic Microbes

There were no significant differences between the activity of chitosan from BSF and that from commercial shrimp on *E. coli* (*p* < 0.05) (Table 3). The activity of BSF and commercial chitosan on *S. aureus*, *C. albicans*, *B. subtilis*, and *P. aeruginosa* had a significant difference. There were no significant differences between the inhibition potential of BSF chitosan and that of the various antibiotics (nitrofurantoin, streptomycin, co-trimoxazole, and gentamycin) against *C. albicans*. No significant differences were observed in the activity of BSF chitosan and antibiotics (nalidixic acid, tetracycline, and gentamycin) against *E. coli*. Other test organisms (*S. aureus, B. subtilis,* and *P. aeruginosa)* were significantly inhibited by BSF chitosan and all other conventional antibiotics. However, no significant differences were observed in the activity of commercial shrimp chitosan and nitrofurantoin, tetracycline, streptomycin, and nalidixic acid against *S*. *aureus*. The activity of commercial shrimp chitosan showed no significant differences as compared to antibiotics (nitrofurantoin, tetracycline, and nalidixic acid) against *B. subtilis*. Amongst conventional antibiotics, gentamycin was sensitive to all of the tested microorganisms, and demonstrated the largest zones of inhibition in each of the test organisms, with *B. subtilis* exhibiting the highest diameter of inhibition. Ampicillin showed no inhibitory effect on any of the tested microbes. *Bacillus subtilis* and *S. aureus* were resistant to sulfathiazole, streptomycin, and co-trimoxazole. Sulfathiazole and co-trimoxazole had no inhibitory effects on *P. aeruginosa*. Conversely, *E. coli* was resistant to ampicillin only, while *C. albicans* was not susceptible to ampicillin or nalidixic acid. These observations are consistent with those reported by the authors of [47], who observed no inhibitory effects of ampicillin on *C. albicans*.

BSF chitosan was found to be more effective against Gram-negative bacteria than Gram-positive bacteria in this study. These results are similar to those of many previous studies [42,58,59,60], which demonstrated higher activity of chitosan against Gram-negative than Gram-positive bacteria. This study, however, differs from some previous studies that demonstrated chitosan as having stronger activity against Gram-positive than Gram-negative bacteria [61,62,63,64]. The killing mechanism of chitosan has not yet been well studied, but several postulates have been proposed thus far. According to [65], the killing mechanism may be attributed to the high concentration of positive charges in the chitosan’s structure, which causes strong electrostatic interactions with negatively charged residues of carbohydrates, proteins, and lipids found in microbial cells, inhibiting bacterial development [65]. It has also been suggested that chitosan alters cell permeability via deposition of chitosan on the pathogen cell surface, resulting in the formation of an impermeable polymeric layer that inhibits nutrient uptake to the cell and changes the metabolite secretions in the extracellular matrix [60,66].

## 4. Conclusions

The work described in this paper demonstrates that chitin can effectively be extracted from the BSF pupal exuviae by co-cultivating *P. aeruginosa* and *B. subtilis* bacteria. Biological extraction gives high yields as compared to chemical extraction and can be used as an alternative method since it is inexpensive and eco-friendly. The FTIR results show that there was a close similarity in the chemical structure and bonding of all of the chitin samples and that they were all of the alpha form (α). This shows that BSF chitin can therefore be used in place of commercial shrimp chitin. Scanning electron microscopy revealed that the surface morphology of both chemical and biological chitin from BSF consisted of both fibers and pores. Chitosan from BSF exhibited activity against all of the tested pathogenic microbes and can therefore be used as a novel drug delivery system for a variety of therapeutic agents for combating antimicrobial-resistant strains. Further studies are needed to demonstrate the mode of action of chitosan on various microbes since the killing mechanism of chitosan remains largely unknown. Therefore, the high economic impact of chitin and chitosan extracted from BSF pupal exuviae should be further exploited for their application in the pharmaceutical, food, cosmetics, textile, wastewater treatment, and agricultural sectors.

## Figures and Tables

**Figure 1 microorganisms-09-02417-f001:**
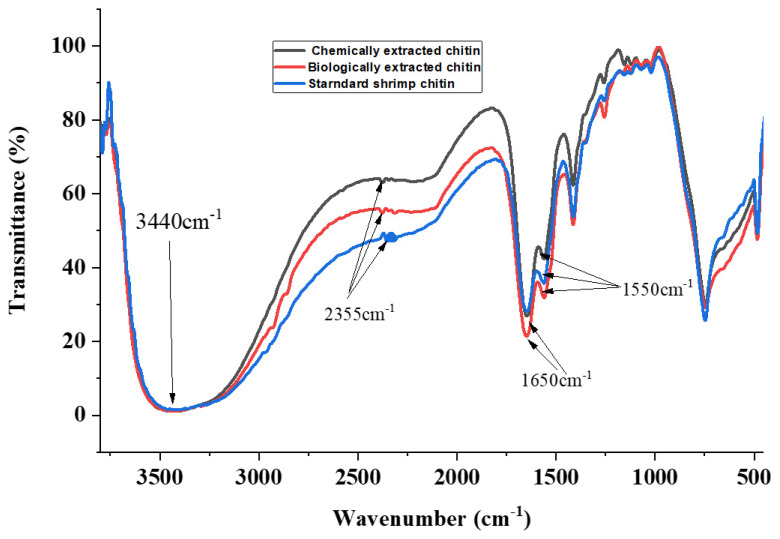
FTIR spectra of chitin extracted from BSF pupae shells via biological and chemical means, compared to commercial standard shrimp chitin (Sigma Aldrich, St. Louis, MI, USA).

**Figure 2 microorganisms-09-02417-f002:**
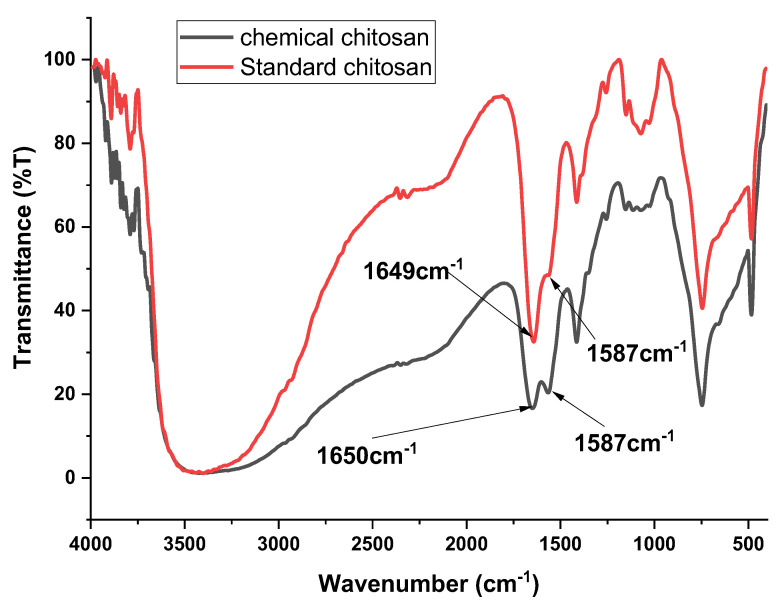
FTIR spectra of chemically extracted chitosan from BSF pupal exuviae, compared to the standard shrimp chitosan (Sigma-Aldrich, St. Louis, MI, USA).

**Figure 3 microorganisms-09-02417-f003:**
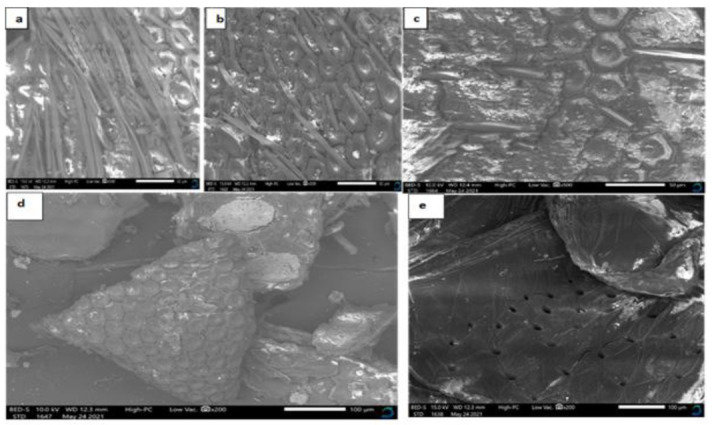
Scanning electron micrographs of (**a**) chitin from BSF pupal shells extracted by *B. subtilis* bacteria, (**b**) chitin from BSF pupal shells extracted by *P. aeruginosa* bacteria, (**c**) chitin from BSF pupal shells extracted by co-cultivation of *B. subtilis* and *P. aeruginosa* bacteria, (**d**) chitin from BSF pupal shells extracted via the use of chemicals, and (**e**) commercial shrimp chitin (Sigma-Aldrich, St. Louis, MI, USA).

**Figure 4 microorganisms-09-02417-f004:**
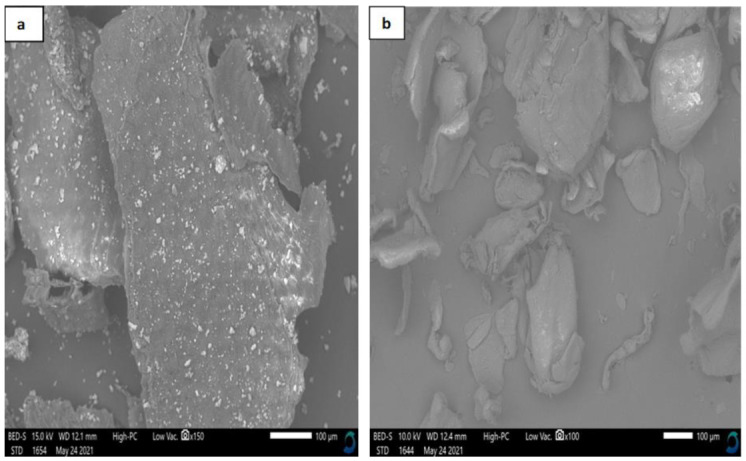
Scanning electron micrographs of (**a**) chitosan from BSF pupal shell chitin obtained via the use of chemicals, and (**b**) commercial shrimp chitosan (Sigma-Aldrich, St. Louis, MI, USA).

**Table 1 microorganisms-09-02417-t001:** Mean chitin yield (g) from biological and chemical extraction from the black soldier fly pupal exuviae.

Chitin Extraction Treatments	Chitin Yield(g) (Mean ± S.D)
*B. subtilis + L. plantarum*	7.78 ± 0.68 ^a^
*P. aeruginosa + L. plantarum*	9.47 ± 0.52 ^b^
*L.plantarum + B. subtilis*	8.76 ± 0.88 ^b^
*L.plantarum+P.aeruginosa*	7.99 ± 1.16 ^a^
*P. aeruginosa + B. subtilis +L. plantarum.*	11.85 ± 1.16 ^c^
Chemical extraction.	10.18 ± 0.42 ^c^
*p*-value	<0.001

Different letters in the same column indicate significant differences (*p* < 0.05). Values represent the mean ± standard deviation.

**Table 2 microorganisms-09-02417-t002:** The mean inhibition zones (mm) of BSF chitosan against selected pathogenic microbes.

Bacteria Species					
Concentration (g/mL)	*C. albicans*	*E. coli*	*S. aureus*	*B. subtilis*	*P. aeruginosa*
0.5	14.33 ± 3.21 ^b^	14.33 ± 3.21 ^b^	13.33 ± 1.53 ^b^	13.67 ± 2.52 ^b^	14.33 ± 0.58 ^b^
1.0	16.67 ± 3.21 ^b^	20.00 ± 4.36 ^c^	16.67 ± 2.31 ^c^	16.67 ± 1.53 ^c^	16.67 ± 1.15 ^c^
2.5	22.00 ± 2.65 ^c^	25.33 ± 2.89 ^c^	21.67 ± 4.93 ^d^	22.67 ± 2.52 ^d^	18.33 ± 1.53 ^d^
5.0	26.00 ± 3.61 ^c^	26.33 ± 2.89 ^c^	23.33 ± 3.79 ^d^	24.67 ± 1.53 ^d^	20.33 ± 1.53 ^d^
1%Acetic acid (+control)	13.33 ± 1.46 ^b^	11.67 ± 1.46 ^b^	11.67 ± 1.46 ^a^	10.00 ± 1.46 ^b^	12.33 ± 1.46 ^b^
Sterile distilled water (-control)	0.00 ± 1.46 ^a^	0.00 ± 1.46 ^a^	0.00 ± 1.46 ^a^	0.00 ± 1.46 ^a^	0.00 ± 1.46 ^a^
*p*-values	0.0081	0.0087	0.0245	0.0006	0.0023

Different letters in the same column indicate significant differences (*p* < 0.05). Values are given as the mean ± standard deviation.

**Table 3 microorganisms-09-02417-t003:** Mean inhibition zones (mm) of BSF chitosan, shrimp chitosan, and conventional antibiotics against selected pathogenic microbes.

Bacteria Species					
Antibiotics	*C. albicans*	*E. coli*	*S. aureus*	*B. subtilis*	*P. aeruginosa*
BSF chitosan	19.75 ± 5.48 ^c^	21.5 ± 5.78 ^d^	18.75 ± 5.07 ^b^	19.42 ± 4.96 ^b^	17.42 ± 2.54 ^b^
Shrimp chitosan	16.58 ± 4.19 ^c^	19.42 ± 5.58 ^d^	15.92 ± 5.32 ^b^	18.17 ± 5.02 ^b^	14.83 ± 3.16 ^b^
Ampicillin (25 mcg)	0.00 ± 1.46 ^a^	0.00 ± 1.46 ^a^	0.00 ± 1.46 ^a^	0.00 ± 1.46 ^a^	0.00 ± 1.46 ^a^
Tetracycline (100 mcg)	25.00 ± 1.46 ^c^	20.00 ± 1.46 ^d^	19.00 ± 1.46 ^b^	24.00 ± 1.46 ^c^	20.00 ± 1.46 ^c^
Nitrofurantoin (200 mcg)	15.00 ± 1.46 ^c^	15.00 ± 1.46 ^d^	19.00 ± 1.46 ^b^	18.00 ± 1.46 ^b^	22.00 ± 1.46 ^d^
Nalidixic acid (30 mcg)	0.00 ± 1.46 ^a^	27.00 ± 1.46 ^e^	21.00 ± 1.46 ^b^	20.00 ± 1.46 ^b^	20.00 ± 1.46 ^d^
Streptomycin (25 mcg)	15.00 ± 1.46 ^c^	15.00 ± 1.46 ^d^	19.00 ± 1.46 ^b^	0.00 ± 1.46 ^a^	15.00 ± 1.46 ^b^
Sulfathiazole (200 mcg)	10.00 ± 1.46 ^b^	10.00 ± 1.46 ^b^	0.00 ± 1.46 ^a^	0.00 ± 1.46 ^a^	0.00 ± 1.46 ^a^
Co-trimoxazole (25 mcg)	17.00 ± 1.46 ^c^	12.00 ± 1.46 ^c^	0.00 ± 1.46 ^a^	0.00 ± 1.46 ^a^	0.00 ± 1.46 ^a^
Gentamycin (10 mcg)	17.00 ± 1.46 ^c^	28.00 ± 1.46 ^e^	29.00 ± 1.46 ^c^	31.00 ± 1.46 ^d^	30.00 ± 1.46 ^e^
*p*-Values	<0.001	<0.001	<0.001	<0.001	<0.001

Different letters in the same column indicate significant differences (*p* < 0.05). Values are given as the mean ± standard deviation.

## Data Availability

All datasets presented in this study are included in the article and can be availed by the authors upon reasonable request.

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
