# Peer review of "Antimicrobial Activity of Chemically and Biologically Treated Chitosan Prepared from Black Soldier Fly (Hermetia illucens) Pupal Shell Waste"

_microorganisms, 2021, doi:10.3390/microorganisms9122417_

Round 1
Reviewer 1 Report
The manuscript deals with interesting topic. The novelty is evident. Obtained results are applicable and are bringing the advancement into the studied subject. The methodology is making sense. The manuscript is relatively well written.
However, there are present some minor issues across the manuscript which decrease its overall quality:
1: There is no need to introduce abbreviations in an abstract part. Moreover, used abbreviation is mentioned later on in an introduction part (line 57). Therefore, abbreviations have to be eliminated from the introduction part.
2: part 2. Materials and methods. This part has to be thoroughly rewritten. There are present several chemical reagents which are of unknown source. Thus, supplier, city and country of the origin has to be given for each reagent.
3: The same has to be done for used equipment. Type, manufacturer, city and country of the origin has to be mentioned for each instrument.
4: lines 267 and 274: use index for (NH2) as in the line 271.
All in all, the manuscript is of enough intertest. Obtained results are valuable. However, minor revision has to be done to reach an adequate quality of the manuscript for publication in Microorganisms.
Author Response
English corrections on the document have been implemented per the reviewers' recommendations.
Line 57; Abbreviation in the abstract removed
Methods; all chemicals and the type of equipment used, the supplier, city, and country of origin have been given for each reagent.
Lines 267 and 274: use index for (NH2) as in line 271, this has been corrected
Reviewer 2 Report
Paper entitled “Antimicrobial Activity of Chemically and Biologically Treated 2 Chitosan Prepared from Black Soldier Fly (Hermetia illucens) Pupal Shell Waste” meets the necessary standards for publication in this journal.
Attention when writing references. They are not unitary.
Please check the entire manuscript carefully for eventual typographical errors. For example, measurement units.
One could make a comparison with other studies.
Final Conclusion: The paper meets the necessary standards for publication.
Author Response
The changes have been done on the methodology section.